# Association between Geriatric Nutrition Risk Index and Skeletal Muscle Mass Index with Bone Mineral Density in Post-Menopausal Women Who Have Undergone Total Thyroidectomy

**DOI:** 10.3390/nu12061683

**Published:** 2020-06-05

**Authors:** Tai-Hua Chiu, Szu-Chia Chen, Hui-Chen Yu, Jui-Sheng Hsu, Ming-Chen Shih, He-Jiun Jiang, Wei-Hao Hsu, Mei-Yueh Lee

**Affiliations:** 1Department of General Medicine, Kaohsiung Medical University Hospital, Kaohsiung 807, Taiwan; tata14080222@gmail.com; 2Division of Nephrology, Department of Internal Medicine, Kaohsiung Medical University Hospital, Kaohsiung Medical University, Kaohsiung 807, Taiwan; scarchenone@yahoo.com.tw; 3Department of Internal Medicine, Kaohsiung Municipal Siaogang Hospital, Kaohsiung Medical University, Kaohsiung 812, Taiwan; my345677@yahoo.com.tw; 4Faculty of Medicine, College of Medicine, Kaohsiung Medical University, Kaohsiung 807, Taiwan; 5Research Center for Environmental Medicine, Kaohsiung Medical University, Kaohsiung 807, Taiwan; 6Department of Medical Imaging, Kaohsiung Medical University Hospital, Kaohsiung 807, Taiwan; l760192@yahoo.com.tw (H.-C.Y.); e3124@ms16.hinet.net (J.-S.H.); stoneshih2007@gmail.com (M.-C.S.); 7Department of Radiology, Faculty of Medicine, College of Medicine, Kaohsiung Medical University, Kaohsiung 807, Taiwan; 8Division of Endocrinology and Metabolism, Department of Internal Medicine, Kaohsiung Medical University Hospital, Kaohsiung Medical University, Kaohsiung 807, Taiwan; 960219kmuh@gmaill.com

**Keywords:** menopausal women, total thyroidectomy, geriatric nutritional risk index, bone mineral density, appendicular skeletal muscle mass

## Abstract

Background: Osteoporosis is highly prevalent in postmenopausal women and may result in fractures and disabilities. Total thyroidectomy has also been associated with loss of bone mass. The aim of this cross-sectional study was to evaluate associations among nutritional status, skeletal muscle index and markers of bone turnover to bone mineral density in postmenopausal women who had undergone total thyroidectomy. Methods: Fifty postmenopausal women who had undergone total thyroidectomy were included. Body composition was measured using dual-energy X-ray absorptiometry (DXA). The Geriatric Nutritional Risk Index (GNRI) was calculated using baseline body weight and serum albumin level. Skeletal muscle mass index was calculated as the appendicular skeletal muscle mass (ASM) divided by the height squared and assessed using DXA. Results. Multivariate stepwise linear regression analysis showed that a low GNRI was significantly associated with low lumbar spine bone mineral density (BMD) and T-score, and that a low ASM/height^2^ was significantly associated with low femoral neck BMD and T-score. A low vitamin D level was significantly associated with low femoral neck BMD and T-score and low total hip BMD and T-score. A high bone alkaline phosphatase (ALP) level was significantly associated with low femoral neck T-score and low total hip BMD and T-score. A low insulin-like growth factor-1 (IGF-1) was significantly associated with low total hip BMD and T-score. Conclusion: In the postmenopausal women who had undergone total thyroidectomy in this study, BMD was positively associated with GNRI, skeletal muscle mass index, and levels of vitamin D and serum IGF-1, and inversely associated with bone ALP level. Nutritional status, skeletal muscle mass index and bone turnover biomarkers can be used to early identify patients with a high risk of osteoporosis in this high-risk group.

## 1. Introduction

Osteoporosis is characterized by low bone mass, compromised bone strength, and structural deterioration of bone tissue. It is a very common progressive skeletal disorder which can cause bone fragility and an increased risk of fractures [1]. The prevalence of osteoporosis in women aged 50 years and over is higher than in men of the same age [2] due to estrogen deficiency and subsequent impairment of the bone turnover cycle [3]. More than 30% of women over the age of 50 years have been reported to have osteoporosis in Taiwan [2]. In United States, the health care burden and cost of osteoporotic fractures is reported to be greater than that of breast cancer, stroke or myocardial infarction [4]. Total thyroidectomy is a procedure to remove all of the thyroid gland when treating thyroid goiter or thyroid tumors. Several studies have described the impact of thyroidectomy on bone mineral density (BMD) and calcium metabolism, and proposed possible mechanisms that may contribute to a reduction in bone mass and osteoporosis after thyroidectomy, including thyroid hormone dysregulation (e.g., calcitonin deficiency) [5] and loss of the protective effect of thyrotropin due to thyroxine treatment [6]. In addition, many studies have reported an association between osteoporosis and postoperative thyrotropin suppression therapy, especially in postmenopausal women who have undergone thyroidectomy [7,8,9].

As a multifactorial systemic disease, many other factors also contribute to osteoporosis. Nutritional status and sarcopenia, the progressive and widespread loss of muscle mass and function, have been associated with a reduction in BMD and osteoporosis [10,11,12]. Several clinical tools have been introduced to evaluate nutritional status, including the Malnutrition Universal Screening Tool [13], Nutrition Risk Score 2002 [14], Malnutrition Screening Tool [15], Mini Nutritional Assessment—Short Form [16], and Geriatric Nutritional Risk Index (GNRI) [17]. Among these tools, the GNRI is a simple index that uses albumin and body mass index (BMI) to assess nutritional status and as an indicator of morbidity and mortality in elderly hospitalized patients. Some studies have reported that the GNRI is an effective prognostic tool for clinical outcomes in patients with cardiovascular disease [18] and in those on hemodialysis [19]. Although sarcopenia is generally thought to be an age-related decrease in skeletal muscle mass accompanied with low muscle function [20], various methods for defining appendicular skeletal muscle mass (ASM) have been reported, including height (Ht) [20,21,22], body weight (Wt) [23] and BMI [24]. Height-adjusted ASM (ASM/Ht^2^, kg/m^2^) has been used to normalize values in studies published by the European Working Group on Sarcopenia in Older People (EWGSOP) [21], International Working Group on Sarcopenia (IWGS) [22] and Asian Working Group for Sarcopenia (AWGS) [20]. In Asian populations, sarcopenia defined by weight-adjusted ASM has been associated with the risk of cardiometabolic disease [25], and sarcopenia defined by height-adjusted ASM has also been closely correlated with more muscular functions than weight-adjusted ASM [26].

Few studies have investigated associations among nutritional status, sarcopenia and osteoporosis in postmenopausal women who have undergone total thyroidectomy. Therefore, in this study, we investigated associations among BMD, as assessed using dual energy X-ray absorptiometry (DXA), the GNRI and ASM/Ht^2^ in postmenopausal women who had undergone total thyroidectomy.

## 2. Subjects and Methods

### 2.1. Study Population

This observational, cross-sectional, controlled study included 50 postmenopausal women who made regular visits to the Outpatient Department of Endocrinology and Metabolism, Kaohsiung Medical University Hospital (KMUH) for treatment and follow-up after complete removal of the thyroid, which included both benign and malignant neoplasms of the thyroid on initial diagnosis. The enrollment period was from 5 July 2019 to 31 December 2019. The exclusion criteria were women with amenorrhea for less than one year and those already being treated for osteoporosis. Diseases that can influence the serum albumin concentration such as liver cirrhosis, acute or chronic inflammations like other types of malignancy, celiac disease, inflammatory bowel disease, chronic kidney disease, systemic lupus erythematous, rheumatoid arthritis, multiple myeloma and any history of organ transplantation were also excluded. Dietary supplements of calcium were allowed. A questionnaire about the risk factors of osteoporosis included the family history with osteoporosis and fractures, amount of diet for calcium intake, medical history of steroids, antiepileptic drug and antacids (only one study population has medical history of antacid use), operation history of the gastrointestinal system, past history of cancer aside from thyroid, personal history of cigarette smoking and alcohol intake, and exercise. This study was approved by the Institutional Review Board of KMUH (KMUHIRB-F(1)- 20190085), and all of the study population read and signed written informed consent. All clinical investigations were carried out according to the principles conveyed in the Declaration of Helsinki.

### 2.2. Biochemical Measurements

Biochemical measurements were performed using standardized methods in the central laboratory of our institution. Serum albumin, aspartate aminotransferase (GOT), alanine aminotransferase (GPT), thyroid stimulating hormone (TSH), free T4, T3, thyroglobulin, thyroglobulin and microsomal antibody, parathyroid hormone, total calcium, urine total protein and creatinine, serum creatinine, follicular stimulating hormone (FSH), estradiol, testosterone, cortisol, insulin growth factor-1 (IGF-1), bone alkaline phosphatase (BALP), vitamin D and bone resorption marker carboxy-terminal collagen crosslinks (CTX) were measured.

### 2.3. BMD and Body Composition Measurements

Body composition was measured using DXA with a Horizon Wi DXA system (Hologic, Waltham, MA, USA). BMD (g/cm^2^) was evaluated in the lumbar spine (L2-L4), femoral neck and total hip. All of the patients were scanned and calculations were performed by one radiologic technologist to minimize variations in measurements. The technician was blinded regarding the study population and the other results of the study. T-scores were used to compare the study subjects to normal sex- and ethnicity-matched individuals with peak bone mass as per the manufacturer’s database [27,28].

### 2.4. Determinants of Skeletal Muscle Mass Index (ASM/ht^2^)

Because overall skeletal muscle mass has an effect on body size, measured skeletal muscle mass needs to be corrected for the body type of the study subject. The correction methods include dividing the skeletal muscle mass of the limbs by the height square (ASM/Ht^2^), weight (ASM/Wt), or body mass index (ASM/BMI). Most current studies, including those published by the EWGSOP, IWGS, and AWGS use ASM/Ht^2^ as assessed by DXA.

### 2.5. Calculation of the GNRI

The GNRI is used to evaluate at-risk elderly medical patients [17], and is calculated using baseline body weight and serum albumin level as follows: GNRI = [14.89 × albumin (g/dL)] + [41.7 × (body weight/ideal body weight)], where body weight/ideal body weight has a value of 1 in patients whose body weight goes beyond the ideal value. We calculated the ideal body weight by measuring the patient’s height and a BMI of 22 kg/m^2^ [29], as previously validated, rather than the Lorentz formula used in the original GNRI equation. BMI was calculated using height and body weight.

### 2.6. Statistical Analysis

Descriptive statistics were reported as percentages, means ± standard deviations, or medians (25th–75th percentile) for non-normally distributed variables. Multiple stepwise linear regression analysis was used to identify the factors associated with BMD and T-score after adjusting for age, a history of thyroid cancer, log-transformed menopausal years, GNRI, ASM/Ht^2^, estimated glomerular filtration rate, total calcium, log-transformed thyroid-stimulating hormone, log-transformed free T4, log-transformed T3, log-transformed parathyroid hormone, log-transformed vitamin D, log-transformed bone alkaline phosphatase (ALP), log-transformed C-terminal telopeptide, log-transformed follicle stimulating hormone, log-transformed estradiol, log-transformed cortisol, log-transformed insulin-like growth factor-1 (IGF-1), log-transformed testosterone, log-transformed thyroglobulin, log-transformed microsomal Ab and log-transformed thyroglobulin Ab. A *p* value < 0.05 was considered to be statistically significant. All statistical analyses were performed using SPSS software for Windows version 19.0 (SPSS Inc., Chicago, IL, USA).

## 3. Results

A total of 50 menopausal women who had undergone total thyroidectomy were enrolled, with a mean age of 61.92 ± 7.77 years. Table 1 shows baseline and DXA characteristics.

### 3.1. Determinants of BMD in the Study Patients

Table 2 shows the determinants of BMD using multivariate stepwise linear regression analysis after adjusting for the factors detailed in the statistical analysis subsection. Old age (per 1 year; unstandardized coefficient β = −0.017; 95% CI, −0.025 to −0.008; *p* < 0.001) and low GNRI (per 1 score; unstandardized coefficient β = 0.009; 95% CI, 0.000 to 0.018; *p* = 0.040) were significantly associated with low lumbar spine BMD. Old age (per 1 year; unstandardized coefficient β = −0.013; 95% CI, −0.018 to −0.008; *p* < 0.001), low ASM/Ht^2^ (per 1 kg/m^2^; unstandardized coefficient β = 0.072; 95% CI, 0.014 to 0.130; *p* = 0.015) and low vitamin D (log per 1 nmol/L; unstandardized coefficient β = 0.271; 95% CI, 0.029 to 0.512; *p* = 0.029) were significantly associated with low femoral neck BMD. In addition, old age (per 1 year; unstandardized coefficient β = −0.011; 95% CI, −0.017 to −0.006; *p* < 0.001), low vitamin D (log per 1 nmol/L; unstandardized coefficient β = 0.285; 95% CI, 0.031 to 0.539; *p* = 0.029), high bone ALP (log per 1 ug/L; unstandardized coefficient β = −0.304; 95% CI, −0.534 to −0.075; *p* = 0.011), and low IGF-1 (log per 1 ng/mL; unstandardized coefficient β, 0.294; 95% CI, 0.004 to 0.584; *p* = 0.047) were significantly associated with low total hip BMD.

### 3.2. Determinants of T-Score in the Study Patients

Table 3 shows the determinants of T-score using multivariate stepwise linear regression analysis after multiple adjustments. Old age (per 1 year; unstandardized coefficient β = −0.122; 95% CI, −0.178 to −0.065; *p* < 0.001) and low GNRI (per 1 score; unstandardized coefficient β = 0.069; 95% CI, 0.010 to 0.127; *p* = 0.022) were significantly associated with low lumbar spine T-score. Old age (per 1 year; unstandardized coefficient β = −0.074; 95% CI, −0.111 to −0.037; *p* < 0.001), low ASM/Ht^2^ (per 1 kg/m^2^; unstandardized coefficient β = 0.557; 95% CI, 0.157 to 0.957; *p* = 0.008), high total calcium (per 1 mg/dL; unstandardized coefficient β = −0.959; 95% CI, −1.782 to −0.137; *p* = 0.023), low vitamin D (log per 1 nmol/L; unstandardized coefficient β = 1.953; 95% CI, 0.287 to 3.618; *p* = 0.023), and high bone ALP (log per 1 ug/L; unstandardized coefficient β = −1.513; 95% CI, −2.932 to −0.094; *p* = 0.037) were significantly associated with low femoral neck T-score. In addition, old age (per 1 year; unstandardized coefficient β = −0.092; 95% CI, −0.135 to −0.049; *p* < 0.001), low vitamin D (log per 1 nmol/L; unstandardized coefficient β = 2.331; 95% CI, 0.330 to 4.331; *p* = 0.023), high bone ALP (log per 1 ug/L; unstandardized coefficient β = −2.438; 95% CI, −4.246 to −0.630; *p* = 0.009), and low IGF-1 (log per 1 ng/mL; unstandardized coefficient β, 2.414; 95% CI, 0.1235 to 4.702; *p* = 0.039) were significantly associated with low total hip T-score.

## 4. Discussion

In the present study, we found significant relationships among nutritional status, sarcopenia and BMD or T-score in postmenopausal women who had undergone total thyroidectomy. There are two important findings in this study. First, a high GNRI score was associated with high L-spine BMD and T-score. An association between BMD and nutritional status has been reported in various patient groups, including those with cardiovascular disease [12], end-stage renal disease [11] and chronic obstructive pulmonary disease [30]. In addition to calcium and vitamin D, protein intake is also associated with bone health. Bonjour et al. reported that low-protein diets are related to hip fractures [31], and Shams-White et al. reported that a higher protein intake may cause less BMD loss in the lumbar spine in a systematic review and meta-analysis [32]. Serum albumin level may reflect nutritional status with regards to protein, and it can be related to bone mass. Coin et al. also reported an association between albumin and BMD in elderly individuals [33]. BMI and body weight can be indicators of energy intake, and the relationship between BMI and BMD has been described in many studies. In an epidemiological study involving 1075 women and 690 men, Nguyen et al. found a positive association between BMD and BMI [34]. The GNRI, which takes both serum albumin and BMI into account, has also been associated with BMD in several recent studies. In a study of 146 Japanese patients, Tokumoto et al. reported that a low GNRI was a risk factor for lower BMD of the femoral neck in patients with rheumatoid arthritis [35]. In addition, Wang et al. reported a positive relationship between GNRI and BMD in Chinese patients with type 2 diabetes mellitus in a study of 225 men and 192 women [36]. In another study of 164 Taiwanese patients, Chen et al. reported that a high GNRI was associated with a higher BMD in patients on hemodialysis [37]. In the present study, we found a positive association between GNRI and L-spine BMD and T-score in postmenopausal women who had undergone total thyroidectomy. This finding reconfirms the association between GNRI and BMD reported in previous studies. As albumin level reflects protein status and is a major component of the GNRI, the effect of protein on bone may help to explain the association between GNRI and BMD. Previous studies have reported that a decrease in protein intake can reduce the level of serum IGF-1 [38], and that IGF-1 can further affect calcium-phosphate metabolism [31] and BMD [39]. In the present study, we also found a positive association between serum IGF-1 level and total hip BMD and T-score in our patient group. It has been well established that IGF-1 and growth hormone (GH) are important regulators of bone metabolism [40]. GH and IGF-1 may affect bone in a synergistic manner, and IGF-1 also has an independent function. In vitro studies have shown that GH can promote the formation of colonies of young pre-chondrocytes, and that IGF-1stimulates cells at a later phase of maturation [41]. Most serum IGF-1 is produced in the liver, and locally produced IGF-1 is derived from bone and other tissue. Both serum and bone-derived IGF-1 contribute to longitudinal bone growth and cortical bone formation, and also the maintenance of bone mass in later stages of life [42]. Many studies have reported a relationship between IGF-1 and bone mass. In an in vivo study, Yakar et al. reported a marked decrease in BMD, periosteal circumference and cortical thickness in IGF-1 gene-disrupted mice [43]. In addition, Seck et al. found that a low serum IGF-1 level was associated with greater femoral bone loss in postmenopausal women [44]. In the large cross-sectional Rancho Bernardo Study of 483 men and 455 postmenopausal women, Barrett-Connor et al. reported a positive relationship between serum IGF-1 concentration and BMD only in women [45]. Moreover, the Longitudinal Aging Study Amsterdam of 627 men and 656 women conducted by van Varsseveld et al. also reported a greater 3-year decrease in total hip BMD in elderly women with lower serum IGF-1 concentrations [46]. In addition to its direct effect on bone, IGF-1 may contribute to bone mass in other ways. The IGF-1 signaling pathway has been shown to be related to skeletal muscle protein synthesis [47]. Serum IGF-1 level has also been related to protein intake [38], which reflects general nutritional status. Furthermore, the complex interaction between sex steroids and IGF-1 may also affect bone mass [48].

The second important finding of this study is that a high ASM/Ht^2^ was associated with high femoral neck BMD and T-score. Emerging evidence has showed the close interaction between bone and muscle, and some studies have even suggested concomitant treatments for both osteoporosis and sarcopenia [49]. Several studies have reported an association between sarcopenia and osteoporosis. In a longitudinal study of 232 patients, Locquet et al. found that a decrease in skeletal muscle mass was associated with a decrease in hip and spine BMD [50]. In addition, Hida et al. reported that the prevalence of sarcopenia was higher among older Japanese women with osteoporotic vertebral fractures than in those who did not have osteoporotic vertebral fractures [51]. Moreover, in a Chinese study of 322 men and 435 women, Hong et al. reported that lower height-adjusted ASM was associated with an increased risk of hip fractures both in elderly men and women [52]. A positive correlation between height-adjusted ASM and BMD has also been reported in Japanese [53] and Korean studies [54]. In the present study, we found that a high ASM/Ht^2^ was associated with higher femoral neck BMD and T-score in postmenopausal women who had undergone total thyroidectomy. Our findings support that sarcopenia defined by height-adjusted ASM is associated with BMD in a high-risk population. The concomitant decrease in bone and muscle may be due to shared regulation such as nutritional, neuronal and endocrine regulation, and also complex muscle-bone cross-talk. The common endocrine regulators of muscle-bone interactions include GH/IGF signaling, calciotropic axis regulation, glucocorticoid receptor signaling and sex steroid signaling [55]. In addition to gravitational loading, bone remodeling is also sensitive to muscular activity-generated internal loads [56]. In addition, muscle can affect bone through local growth factors or myokines, and bone may affect muscle through osteokines or clastokines [55]. 

Another important finding of this study is that a high serum vitamin D level was associated with high femoral neck BMD and T-score, and high total hip BMD and T-score. It is widely known that vitamin D plays a major role in mineral metabolism and bone health through endocrine effects on bone, parathyroid glands, intestines and kidneys [57], and that a low vitamin D status can contribute to bone loss and low BMD. A positive association between serum vitamin D level and BMD has been reported in many studies, mostly in Western populations. In the Longitudinal Aging Study Amsterdam conducted in the Netherlands, Kuchuk et al. found that serum 25-hydroxyvitamin D (25(OH)D) was positively associated with BMD when its concentration was not above about 50–60 nmol/L [58]. In another study of 7441 postmenopausal women from 29 countries, a significant association was found between serum 25(OH)D and BMD [59]. In a Swedish study of elderly Scandinavians, Melin et al. reported that femoral neck BMD was significantly and positively associated with 25(OH)D [60]. In an Asian population, Chailurkit et al. reported a positive relationship between serum 25(OH)D level and femoral neck BMD [61]. However, many inconsistent findings have also been reported. In a review article, Man et al. analyzed 11 studies, only five of which showed a positive association between vitamin D status and BMD in middle-aged and older Chinese individuals [62]. In addition, Kota et al. found no direct relationship between serum 25(OH)D levels and BMD in an Indian population [63]. In the present study, we found that a high vitamin D level was associated with high femoral neck BMD and T-score in postmenopausal women who had undergone total thyroidectomy. Vitamin D affects skeletal mineralization mainly by regulating intestinal calcium absorption [64]. In addition, several studies have also shown that vitamin D promotes calcium and phosphate transport and protein synthesis in muscle [65], thereby influencing skeletal muscle mass and muscle strength [66]. This may suggest that vitamin D can be used in this population to improve osteoporosis. 

The assessment of bone health and diagnosis of osteoporosis are currently mainly based on DXA or other imaging techniques. However, these techniques have intrinsic limitations, such as the inability to evaluate bone microarchitecture, and thus they provide limited information about bone strength. Several bone turnover markers have been identified that can provide supplementary information about bone loss. Bone ALP is an enzyme expressed on the cell surface of osteoblasts, and it is considered to be involved in bone formation or mineralization. It is one of the most widely used indicators of bone metabolism, and it has been shown to be elevated in bone metabolic disorders including osteoporosis and Paget’s disease of bone [67]. Many studies have reported an inverse association between bone ALP and BMD. In a Chinese study of 4197 patient, Chen et al. reported that a high bone ALP level was associated with low BMD in patients with and without diabetes [68]. In addition, Bergman et al. reported that bone ALP can predict BMD in patients with end-stage renal disease [69]. Moreover, Nakamura et al. reported that bone ALP was negatively correlated with lumbar spine BMD in Japanese postmenopausal osteoporotic women receiving denosumab treatment [70]. Furthermore, a systemic review conducted by Biver et al. also reported moderate and negative correlations between bone ALP and BMD mainly in postmenopausal women [71]. In the present study, we found that high bone ALP was associated with low femoral neck T-score and low total hip BMD and T-score in postmenopausal women who had undergone total thyroidectomy. Our results are in accordance with most previous findings, suggesting that high bone ALP is related to increased bone turnover in postmenopausal women. 

There are several limitations to this study. First, this is a cross-sectional study, so we cannot evaluate longitudinal relationships between these parameter and BMD or T-score. Second, we only obtained data on serum IGF-1 level and we did not have data on IGF-binding protein level or local bone IGF-1, which are also related to bone physiology [40]. Third, as some patients were treated for neoplasms of the thyroid gland in our study, we cannot exclude the possibility that the observed findings may partly be explained by other variables such as previous treatment or medication. Fourth, several parameters used in this study were age dependent; however, we were not able to include control potential confounder with a group of healthy postmenopausal women of the same group. Lastly, because the relatively small number of enrolled patients, it is not enough to use statistical tests draw reliable conclusions. Follow-up large-scale studies are needed to confirm our results.

## 5. Conclusions

This study of postmenopausal women with younger age who had undergone total thyroidectomy demonstrated a positive association between BMD and nutritional status measured using the GNRI, skeletal muscle mass index, vitamin D level and serum IGF-1 level, and an inverse relationship between serum bone ALP level and BMD. Our findings suggest that in this specific patient group of postmenopausal women who have undergone total thyroidectomy, it is possible to identify those at a high risk of osteoporosis not only through BMD but also through parameters of nutritional status, skeletal muscle mass index and bone turnover biomarkers.

## Figures and Tables

**Table 1 nutrients-12-01683-t001:** Baseline and dual energy X-ray absorptiometry (DXA) characteristics.

Characteristics	All Patients (*n* = 50)
Age (year)	61.92 ± 7.77
Papillary type of thyroid cancer (%)	62.0
Menopausal years (year)	12.00 (8.25–17.50)
GNRI (score)	112.68 ± 7.38
Height (cm)	156.48 ± 5.52
Weight (kg)	60.18 ± 9.12
BMI (kg/m^2^)	24.55 ± 3.35
Time after thyroidectomy (years)	5.00 (1.00–14.00)
Total Levothyroxine dose (mcg)	14400 (4200–43200)
DXA Parameters	
Lumbar spine BMD (g/cm^2^)	0.99 ± 0.26
T score	−1.40 ± 1.75
Femoral neck BMD (g/cm^2^)	0.81 ± 0.17
T score	−1.62 ± 1.23
Total hip BMD (g/cm^2^)	0.89 ± 0.17
T score	−0.94 ± 1.40
Body composition	
ASM/height^2^ (kg/m^2^)	6.12 ± 0.64
Lean mass (trunk, %)	48.06 ± 1.72
Lean mass (upper and lower extremity, %)	42.83 ± 1.98
Fat (trunk, %)	54.50 ± 4.72
Fat (upper and lower extremity, %)	41.27 ± 4.53
Laboratory parameters	
Albumin (g/dL)	4.44 ± 0.23
eGFR (mL/min/1.73 m^2^)	86.17 ± 16.17
Total calcium (mg/dL)	8.92 ± 0.37
TSH (mU/L)	0.16 (0.03–1.74)
Free T4 (ug/dL)	1.68 (1.44–2.00)
T3 (ng/mL)	74.80 (66.60–90.13)
PTH (pg/mL)	28.18 (21.95–33.07)
Vitamin D (nmol/L)	25.80 (21.20–31.85)
Bone ALP (ug/L)	13.90 (10.90–18.00)
CTx (ng/mL)	0.27 (0.17–0.35)
FSH (mIU/mL)	41.57 (27.48–63.94)
Estradiol (pg/mL)	19.93 (16.43–26.68)
Cortisol (ug/dL)	10.63 (8.63–12.40)
IGF–1 (ng/mL)	113.94 (92.88–154.64)
Testosterone (ng/dL)	34.00 (24.80–44.20)
Thyroglobulin (IU/mL)	0.16 (0.16–0.16)
Microsomal Ab (IU/mL)	13.10 (10.00–22.20)
Thyroglobulin Ab (IU/mL)	20.00 (20.00–20.00)

Abbreviations. GNRI, geriatric nutrition risk index; BMI, body mass index; DXA, dual-energy X-ray absorptiometry; BMD, bone mineral density; ASM, appendicular skeletal muscle; eGFR, estimated glomerular filtration rate; TSH, Thyroid-stimulating hormone; PTH, parathyroid hormone; ALP, Alkaline Phosphatase; CTx, C-terminal telopeptide; FSH, follicle-stimulating hormone; IGF-1, Insulin-like growth factor-1; Ab, antibody.

**Table 2 nutrients-12-01683-t002:** Determinants of bone mineral density (BMD) using multivariable stepwise linear regression analysis.

BMD	Multivariate (Stepwise)	
	Unstandardized coefficient β (95% CI)	*p*
Lumbar spine BMD		
Age (per 1 year)	−0.017 (−0.025, −0.008)	<0.001
GNRI (per 1 score)	0.009 (0.000, 0.018)	0.040
Femoral neck BMD		
Age (per 1 year)	−0.013 (−0.018, −0.008)	<0.001
ASM/height^2^ (per 1 kg/m^2^)	0.072 (0.014, 0.130)	0.015
Vitamin D (log per 1 nmol/L)	0.271 (0.029, 0.512)	0.029
Total hip BMD		
Age (per 1 year)	−0.011 (−0.017, −0.006)	<0.001
Vitamin D (log per 1 nmol/L)	0.285 (0.031, 0.539)	0.029
Bone ALP (log per 1 ug/L)	−0.304 (−0.534, −0.075)	0.011
IGF-1 (log per 1 ng/mL)	0.294 (0.004, 0.584)	0.047

Adjusting for age, a history of thyroid cancer, log-transformed menopausal years, GNRI, ASM/height^2^, eGFR, total calcium, log-transformed TSH, log-transformed free T4, log-transformed T3, log-transformed PTH, log-transformed vitamin D, log-transformed bone ALP, log-transformed CTx, log-transformed FSH, log-transformed estradiol, log-transformed cortisol, log-transformed IGF-1, log-transformed testosterone, log-transformed thyroglobulin, log-transformed microsomal Ab and log-transformed thyroglobulin Ab. Abbreviations are same as Table 1.

**Table 3 nutrients-12-01683-t003:** Determinants of T-score using multivariable stepwise linear regression analysis.

T-Score	Multivariate (Stepwise)	
	Unstandardized coefficient β (95% CI)	*p*
Lumbar spine T-score		
Age (per 1 year)	−0.122 (−0.178, −0.065)	<0.001
GNRI (per 1 score)	0.069 (0.010, 0.127)	0.022
Femoral neck T-score		
Age (per 1 year)	−0.074 (−0.111, −0.037)	<0.001
ASM/height^2^ (per 1 kg/m^2^)	0.557 (0.157, 0.957)	0.008
Total calcium (per 1 mg/dL)	−0.959 (−1.782, −0.137)	0.023
Vitamin D (log per 1 nmol/L)	1.953 (0.287, 3.618)	0.023
Bone ALP (log per 1 ug/L)	−1.513 (−2.932, −0.094)	0.037
Total hip T-score		
Age (per 1 year)	−0.092 (−0.135, −0.049)	<0.001
Vitamin D (log per 1 nmol/L)	2.331 (0.330, 4.331)	0.023
Bone ALP (log per 1 ug/L)	−2.438 (−4.246, −0.630)	0.009
IGF-1 (log per 1 ng/mL)	2.414 (0.125, 4.702)	0.039

Adjusting for age, a history of thyroid cancer, log-transformed menopausal years, GNRI, ASM/height^2^, eGFR, total calcium, log-transformed TSH, log-transformed free T4, log-transformed T3, log-transformed PTH, log-transformed vitamin D, log-transformed bone ALP, log-transformed CTx, log-transformed FSH, log-transformed estradiol, log-transformed cortisol, log-transformed IGF-1, log-transformed testosterone, log-transformed thyroglobulin, log-transformed microsomal Ab and log-transformed thyroglobulin Ab. Abbreviations are same as Table 1.

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
