# Peer review of "Association between Geriatric Nutrition Risk Index and Skeletal Muscle Mass Index with Bone Mineral Density in Post-Menopausal Women Who Have Undergone Total Thyroidectomy"

_nutrients, 2020, doi:10.3390/nu12061683_

Round 1
Reviewer 1 Report
Manuscripts should be published on the basis of good scientific quality. This work has several flaws, starting with the research design. The study group consists of 50 postmenopausal women after a total thyroidectomy, which when divided into two subgroups according to the T-score values, gives a sample size of 12 and 38 respectively. It's not enough to use specific statistical tests and to draw reliable conclusions. Several parameters used in this study are age-dependent. It is necessary to control potential confounders with a group of healthy women of the same age.
The study group should be better characterized depending on the type of cancer, time after the thyroid gland resection, treatment and medication. The methods do not describe the laboratory tests used. They are known only from the statistical section. The cut-off values for GNRI have not been described. How many women in this study had GNRI below 98, which means that they are at risk of complications?
Table 1 shows incorrect data for IGF-1 in the second column. Please also check the results for thyroglobulin. The results are the same in both columns. Strangely, p-value is 0.056. The same is for thyroglobulin Ab with p=0.488.
Tables 2 and 3 show that in all cases age is a determining factor. It is well-known that BMD and vitamin D tend to be lower in older people, while bone ALP tends to be higher compared to younger age groups. These parameters as well as nutrition status and muscle mass deteriorate with age. Therefore, it is difficult to see any novelty in this study.
The discussion is multi-faceted, but it does not compensate for the shortcomings from other parts of the manuscript.
The authors conclude that nutritional status, skeletal muscle mass index and bone turnover markers can be used to identify patients with a high risk of osteoporosis among postmenopausal women after thyroidectomy. I don't see how these results can be put into practice in relation to women after thyroidectomy. Couldn’t the same conclusions be drawn for women after menopause without thyroid surgery?
Technical comments. Running title isn’t informative. No keywords were given either.
Author Response
Reviewer 1
- Manuscripts should be published on the basis of good scientific quality. This work has several flaws, starting with the research design. The study group consists of 50 postmenopausal women after a total thyroidectomy, which when divided into two subgroups according to the T-score values, gives a sample size of 12 and 38 respectively. It's not enough to use specific statistical tests and to draw reliable conclusions. Several parameters used in this study are age-dependent. It is necessary to control potential confounders with a group of healthy women of the same age.
Ans: Thank you for your comments.
- Because the relatively small number of enrolled patients, it's not enough to use statistical tests draw reliable conclusions. We have added this issue in the Limitation.
- Lastly, because the relatively small number of enrolled patients, it's not enough to use statistical tests draw reliable conclusions. Follow-up large-scale studies are needed to confirm our results. (Line 341-342)
- We were initially to study post total thyroidectomy postmenopausal women with non-thyroid cancer as control versus thyroid cancer, but due to relatively small number of enrolled patients, we were not able to reach a significant outcome with these two groups. Healthy postmenopausal women not usually visited our clinic, therefore we were not able to enrolled this type of patient as control group.
- The study group should be better characterized depending on the type of cancer, time after the thyroid gland resection, treatment and medication. The methods do not describe the laboratory tests used. They are known only from the statistical section. The cut-off values for GNRI have not been described. How many women in this study had GNRI below 98, which means that they are at risk of complications?
Ans: Thank you for your precious comments.
- In our country, the highest incidence of thyroid cancer is papillary type, and all of our study population positive for thyroid cancer were all papillary type. We indicated the papillary type of thyroid cancer in Table 1 at revised one. We have added time after thyroidectomy in years, and total levothyroxine dose in Table 1 as below. No patients use steroids or antiepileptic drugs, and just one patient use antacids, which was now indicated at the methods section. Therefore, we did not put them into Table 1. No patients had GNRI below 98.
|
Characteristics |
T-score ≥ -1.0 at each site (n = 12) |
T-score < -1 at any site (n = 38) |
p |
|
Time after thyroidectomy (years) |
6 (2–13.75) |
5 (1–14.25) |
0.895 |
|
Total Levothyroixine dose (mcg) |
12600 (6862.5-41400) |
14400 (4162.5-46800) |
0.968 |
- Details about biochemical measurements were now described at method section on revised version.
- Questionnaire about the risk factors of osteoporosis was inquired included the family history with osteoporosis and fractures, amount of diet for calcium intake, medical history of steroids, antiepileptic drug and antacids ( only one study population has medical history of antacid use), operation history of gastrointestinal system, past history of cancer aside from thyroid, personal history of cigarettes smoking and alcohol intake and exercise. (Line 106-110)
- Biochemical measurements
Biochemical measurements were performed using standardized methods in the central laboratory of our institution. Serum albumin, aspartate aminotransferase (GOT), alanine aminotransferase (GPT), thyroid stimulating hormone (TSH), free T4, T3, thyroglobulin, thyroglobulin and microsomal antibody, parathyroid hormone, total calcium, urine total protein and creatinine, serum creatinine , follicular stimulating hormone (FSH), estradiol, testosterone , cortisol , insulin growth factor-1 (IGF-1), bone alkaline phosphatase (BALP), vitamin D and bone resorption marker carboxy-terminal collagen crosslinks (CTX) were measured. (Line 116-123)
- Table 1 shows incorrect data for IGF-1 in the second column. Please also check the results for thyroglobulin. The results are the same in both columns. Strangely, p-value is 0.056. The same is for thyroglobulin Ab with p=0.488.
Ans: Thank you for your careful correction. The data of IGF-1 at second column was incorrect. We have corrected the error. Other statistical data is correct after check. After double check with the data of thyroglobulin, thyroglobulin and microsomal Ab, no technical error on data input was done. Since all of our study population was already underwent total thyroidectomy and were non autoimmune thyroiditis patients, so the data of thyroglobulin (<0.16 IU/mL) and thyroglobulin Ab ( <20 IU/mL) will be as low as not detected level.
|
Characteristics |
T-score ≥ -1.0 at each site (n = 12) |
T-score < -1 at any site (n = 38) |
p |
|
IGF-1 (ng/mL) |
130.75 (106.97-156.99) |
107.8 (91.0 - 154.6) |
0.851 |
|
Testosterone (ng/dl) |
37.60 (26.53-42.63) |
32.8 (24.15-44.80) |
0.507 |
|
Thyroglobulin (IU/mL) |
0.16 (0.16-0.16) |
0.16 (0.16-0.16) |
0.056 |
|
Thyroglobulin Ab (IU/mL) |
20.00 (20.00-20.00) |
20.00 (20.00-20.00) |
0.488 |
- (1) Tables 2 and 3 show that in all cases age is a determining factor. It is well-known that BMD and vitamin D tend to be lower in older people, while bone ALP tends to be higher compared to younger age groups. These parameters as well as nutrition status and muscle mass deteriorate with age. Therefore, it is difficult to see any novelty in this study.
- The discussion is multi-faceted, but it does not compensate for the shortcomings from other parts of the manuscript.
- The authors conclude that nutritional status, skeletal muscle mass index and bone turnover markers can be used to identify patients with a high risk of osteoporosis among postmenopausal women after thyroidectomy. I don't see how these results can be put into practice in relation to women after thyroidectomy. Couldn’t the same conclusions be drawn for women after menopause without thyroid surgery?
Ans. We agree with what you mentioned above. But we think the novelty of our study was happened in the younger age (mean age of 54 vs 64) of postmenopausal women with long term supplements of levothyroixine, an osteoporotic agent. These 2 risk factors together in younger age post menopausal women, instead of older than 65 years old postmenopausal elderly women, might accelerate the deterioration of BMD, Vit. D and bone ALP, which might affect the nutrition status and muscle mass that was independent from age. In this study, we aim to emphasize the early detection and prevention of osteoporosis and sarcopenia in the population of postmenopausal women which needs long term supplement of levothyroxine disregard of the age.
- Technical comments. Running title isn’t informative. No keywords were given either.
Ans. Thank you for your suggestion. However, running title was limited by word count. In addition, keywords were listed below the abstract, on line 51 and 52.
- Keywords: menopausal women; total thyroidectomy; geriatric nutritional risk index; bone marrow density; appendicular skeletal muscle mass (Line 51-52)

Reviewer 2 Report
Overall Comments
In the present article “Association Between Geriatric Nutrition Risk Index and Skeletal Muscle Mass Index with Bone Marrow Density in Post-Menopausal Women Who Have Undergone Total Thyroidectomy” the authors were able to identify markers with which postmenopausal women after thyroidectomy with a high risk of osteoporosis can be identified. Based on the methods and the outcomes, the results are relevant for this field of research. The article is well-written, but my comments below should be considered.
Specific Comments to the Authors
Line Comment
58 Please use the original source
61 Singer et al. does not write about the prevalence. Also, the incidence of osteoporosis fractures is not greater than that for breast cancer, stroke and myocardial infarction combined. Furthermore, it would be good to mention that this investigation took place in the USA.
112 Please add reference.
114 Please add reference for T-Score.
116 I do not understand the whole paragraph. Was ASM calculated in relation to height, weight and BMI?
188 Is there any specific literature that the serum albumin level depends on the protein intake?
205 Please use IGF-1 instead of IGF-I.
277 Please add reference.
Author Response
Reviewer 2
In the present article “Association Between Geriatric Nutrition Risk Index and Skeletal Muscle Mass Index with Bone Marrow Density in Post-Menopausal Women Who Have Undergone Total Thyroidectomy” the authors were able to identify markers with which postmenopausal women after thyroidectomy with a high risk of osteoporosis can be identified. Based on the methods and the outcomes, the results are relevant for this field of research. The article is well-written, but my comments below should be considered.
Specific Comments to the Authors
Line Comment
- 58 Please use the original source
Ans. Thank you for your careful correction, we have replaced the reference with the original source. We have also replaced the term “postmenopausal women” with “women aged 50 years and over” to be in line with original article. (Line57-58)
- 61 Singer et al. does not write about the prevalence. Also, the incidence of osteoporosis fractures is not greater than that for breast cancer, stroke and myocardial infarction combined. Furthermore, it would be good to mention that this investigation took place in the USA.
Ans. Thank you for your careful review. The finding of Singer et al. was that hospitalization burden and cost is greater than that of breast cancer, stroke or myocardial infarction. We have revised the text in introduction.
- In United States, the health care burden and cost of osteoporotic fractures is reported to be greater than that of breast cancer, stroke or myocardial infarction. (Line 61-62)
- 112 Please add reference
Ans. Thank you for your careful review. We have added the reference for this text. According to International Society for Clinical Densitometry (ISCD), the reference standard for T-score calculation is from NHANES III database (female, white). In addition, because no significant difference was noted in fracture rate between Caucasians and Taiwanese, the guidelines in Taiwan also suggested that the WHO criteria for osteoporotic diagnosis, which was designed for Caucasian, could be also used in Taiwanese population. (Line 131)
- 114 Please add reference for T-Score.
Ans. Thank you for your careful review. We have added the reference for T-score.(Line 133)
- 116 I do not understand the whole paragraph. Was ASM calculated in relation to height, weight and BMI?
Ans. Thank you for your question. The original data of ASM was measured with DXA. Because overall skeletal muscle mass is associated with body size (i.e. individual who is taller or heavier may have much skeletal muscle), the originally measured data of ASM needs to be corrected for the body size of the study subject. ASM/ height square (ASM/Ht2) is widely used in most current studies, and we also adapted ASM/Ht2 as the correction method of ASM in our study. (Line 135-139)
- 188 Is there any specific literature that the serum albumin level depends on the protein intake?
Ans. Thank you for raising this question. A few studies have discussed this topic. In a study of 16 old men and 50 old women (aged 79 and 82 years respectively) in England, albumin levels was reported to positively correlated to protein intake in women, and albumin levels positive correlated to protein to calorie ratios in both women and men [A]. In another study of 7715 Japanese subjects, a weak positive association between animal protein intakes and serum albumin levels was found [B]. In addition to serum albumin level, dietary protein intake was also reported to be related to albumin synthesis rate [C].
[A] MacLennan, W.J.; Martin, P.; Mason, B.J. Protein intake and serum albumin levels in the elderly. Gerontology 1977, 23, 360-367.
[B] Watanabe, M.; Higashiyama, A.; Kokubo, Y.; Ono, Y.; Okayama, A.; Okamura, T.; Group, N.D.R. Protein intakes and serum albumin levels in a japanese general population: Nippon data90. J Epidemiol 2010, 20 Suppl 3, S531-536.
[C] Thalacker-Mercer, A.E.; Campbell, W.W. Dietary protein intake affects albumin fractional synthesis rate in younger and older adults equally. Nutr Rev 2008, 66, 91-95.
- 205 Please use IGF-1 instead of IGF-I.
Thank you for your careful correction. We have revised the text.
- 277 Please add reference.( vit D and endocrine)
Ans. Thank you for your advice. We have added reference about Vitamin D and its endocrine effects.

Reviewer 3 Report
The authors investigated the relation between nutritional and endocrine variables with bone mineral density and T-scores at different locations in postmenopausal women after thyroidectomy. They found positive correlations with the geriatric nutritional risk index (GNRI), muscle mass (as ASM/height2), vitamin D and IGF-1 and a negative correlation with alkaline phosphatase.
I have only minor comments:
- Page 3/Assessment of bone mineral density: The measurements and calculations were performed by the same technician. Was the technician blinded regarding the patients and the other results of the study?
- Page 4/GNRI: The serum albumin concentration can also be influenced by chronic liver disease (liver cirrhosis) and acute or chronic inflammations (negative acute phase protein). Were such patients excluded from the study?
- Page 4/normal BMI: The authors used a BMI of 22 kg/m2 as the normal value. 22 kg/m2 appears to be low; please comment.
- Table 1: IGF1 appears to be different between the two groups, please, check. The values for thyroglobulin and thyroglobulin AB are identical; this seems to be a copy paste error. Please, check.
- Discussion, line 204: BMD or T-score.
Author Response
Reviewer 3
The authors investigated the relation between nutritional and endocrine variables with bone mineral density and T-scores at different locations in postmenopausal women after thyroidectomy. They found positive correlations with the geriatric nutritional risk index (GNRI), muscle mass (as ASM/height2), vitamin D and IGF-1 and a negative correlation with alkaline phosphatase.
I have only minor comments:
- Page 3/Assessment of bone mineral density: The measurements and calculations were performed by the same technician. Was the technician blinded regarding the patients and the other results of the study?
Ans. Yes, the role of the technician in this study was purely performing the DEXA and body composition test. And this statement was already added in the method section (BMD and body composition determinants).
*Technician was blinded regarding the study population and the other results of the study. (Line 128-129)
- Page 4/GNRI: The serum albumin concentration can also be influenced by chronic liver disease (liver cirrhosis) and acute or chronic inflammations (negative acute phase protein). Were such patients excluded from the study?
Ans. Yes, this statement was already added in the method section (study population).
* Diseases that can influence the serum albumin concentration such as liver cirrhosis, acute or chronic inflammations like other types of malignancy, celiac disease, inflammatory bowel disease, chronic kidney disease, systemic lupus erythematous, rheumatoid arthritis, multiple myeloma and any history of organ transplantation were also excluded. (Line 102-105)
- Page 4/normal BMI: The authors used a BMI of 22 kg/m2 as the normal value. 22 kg/m2appears to be low; please comment.
Ans. Thank you for raising this point. The main reason we use BMI of 22 kg/m2 to calculated the ideal body weight in GNRI is its validity [E]. Besides, according to WHO expert consultation, in different Asian populations, the BMI cut-off point for observed risk varies from 22 kg/m2 to 25 kg/m2 [A]. In addition, considering the metabolic risk factor such as diabetes, dyslipidemia and hypertension, the optimal cutoffs for women aged 40 and over in Taiwan was 22.4 kg/m2 [B]. In another Taiwanese study, the mean of BMI in normal weight group(18.5≦BMI<24) is 21.5 kg/m2 [C]. With the same BMI, Asian tends to have higher body adiposity than Caucasians [D]. Therefore, in Asian and Taiwanese population, we suggest to retain that BMI of 22 kg/m2 as the normal value for postmenopausal women.
[A] Consultation, W.H.O.E. Appropriate body-mass index for asian populations and its implications for policy and intervention strategies. Lancet 2004, 363, 157-163.
[B] Hsu, H.S.; Liu, C.S.; Pi-Sunyer, F.X.; Lin, C.H.; Li, C.I.; Lin, C.C.; Li, T.C.; Lin, W.Y. The associations of different measurements of obesity with cardiovascular risk factors in chinese. Eur J Clin Invest 2011, 41, 393-404.
[C] Chang, H.C.; Yang, H.C.; Chang, H.Y.; Yeh, C.J.; Chen, H.H.; Huang, K.C.; Pan, W.H. Morbid obesity in taiwan: Prevalence, trends, associated social demographics, and lifestyle factors. PLoS One 2017, 12, e0169577.
[D] Chang, C.J.; Wu, C.H.; Chang, C.S.; Yao, W.J.; Yang, Y.C.; Wu, J.S.; Lu, F.H. Low body mass index but high percent body fat in taiwanese subjects: Implications of obesity cutoffs. Int J Obes Relat Metab Disord 2003, 27, 253-259.
[E] Shah, B.; Sucher, K.; Hollenbeck, C.B. Comparison of ideal body weight equations and published height-weight tables with body mass index tables for healthy adults in the united states. Nutr Clin Pract 2006, 21, 312-319.
- Table 1: IGF1 appears to be different between the two groups, please, check. The values for thyroglobulin and thyroglobulin AB are identical; this seems to be a copy paste error. Please, check.
Ans: Thank you for your careful correction. The data of IGF-1 at second column was incorrect. We have corrected the error. Other statistical data is correct after check.
|
Characteristics |
T-score ≥ -1.0 at each site (n = 12) |
T-score < -1 at any site (n = 38) |
p |
|
IGF-1 (ng/mL) |
130.75 (106.97-156.99) |
107.8 (91.0 - 154.6) |
0.851 |
|
Thyroglobulin (IU/mL) |
0.16 (0.16-0.16) |
0.16 (0.16-0.16) |
0.056 |
|
Thyroglobulin Ab (IU/mL) |
20.00 (20.00-20.00) |
20.00 (20.00-20.00) |
0.488 |
- Discussion, line 204: BMD or T-score.
Ans. Thank you for your careful correction. We have revised the text.

Reviewer 4 Report
ABSTRACT: "bone marrow density" should be written bone mineral density
THE AUTHORS PROPOSE TO STUDY THE ASSOCIATION BETWEEN NUTRITIONAL STATE, SARCOPENIA AND OSTEOPOROSIS IN MENOPAUSE. TWO GROUPS OF WOMEN ARE SELECTED ON THE BASIS OF T-score> or <of -1. THIS T-score DOES NOT DISTINGUISH BETWEEN WOMEN WITH AND WITHOUT OSTEOPOROSIS. THIS IS NOT IN LINE WITH THE PURPOSE OF THE STUDY.
LINE 95. IT IS A VERY UNCOMMON DEFINITION. THE WAY THE STUDY WAS CONDUCTED NEED TO BE DETAILED.
LINE 110.... T-scores were used to compare the study subjects to normalsex ... The meaning of this sentence is unclear. What women are matched by ethnicity?
How many subject entered the multivariate model?
Line 120: SOMETHING ABOUT LABORATORY TESTS YOU USED SHOULD BE WRITTEN.
Line 115: A T-score <-1 IS NOT NECESSARY abnormal. IT MAY BE NORMAL FOR AGE. Anyway It does not correspondsto the diagnosis of osteoporosis.
Table 1: weight and height are missing.
table 1: FEMORAL neck is g/cm2.
Table 3: THE T-SCORE IS A MEASURE DERIVED FROM THE BMD. GENERALLY THE BMD IS ONLY USED FOR THE STATISTICAL ANALYSIS OF THE DATA BECAUSE THERE SHOULD NOT BE STATISTICAL DIFFERANCES BETWEEN THE TWO MEASURES. HOW DO YOU EXPLAIN THE DIFFERENCE OF CORRELATION WITH TOTAL CALCIUM AND ASM BETWEEN BMD AND T-SCORE IN FEMORAL SITES?
Line 261: You do not have selected women with osteoporosis. So your data suggest that sarcopenia is associated with BMD.
Line 230: BMD?
Women of the groups with "normal and abnormal" T-scores do not differ in nutritional status and muscle mass. Don't you think this contradicts the association of BMD with nutritional status and muscle mass? what is the usefulness of this comparison for the purpose of what the study wants to demonstrate?
Author Response
Reviewer 4
- ABSTRACT: "bone marrow density" should be written bone mineral density
Ans: Thank you for your correction. We have revised.
- THE AUTHORS PROPOSE TO STUDY THE ASSOCIATION BETWEEN NUTRITIONAL STATE, SARCOPENIA AND OSTEOPOROSIS IN MENOPAUSE. TWO GROUPS OF WOMEN ARE SELECTED ON THE BASIS OF T-score> or <of -1. THIS T-score DOES NOT DISTINGUISH BETWEEN WOMEN WITH AND WITHOUT OSTEOPOROSIS. THIS IS NOT IN LINE WITH THE PURPOSE OF THE STUDY.
Ans: Thank you for raising this point. T-score is the standard deviation units used to evaluate the BMD status in relation to the young population. We agree that T-score < -1 may not necessarily corresponds to abnormal, but the diagnosis criteria of osteoporosis or low bone mass/osteopenia using T-score as reference is suggested to be used in postmenopausal women and in men aged 50 year and over. We also agree your opinion that T-score < -1 does not equal to diagnosis of osteoporosis. However, according to WHO criteria, T-score < -1 and > -2.5 represents low bone mass/osteopenia, which is also an abnormal status and the hallmark of progression to osteoporosis. Besides, all of our study individuals are postmenopausal women that eligible to use T-score as diagnosis reference. Thus we would like to retain T-score < -1 as cut-off point for “abnormal” in our study. References: Reference [27],[28],[29] in our manuscript
- LINE 95. IT IS A VERY UNCOMMON DEFINITION. THE WAY THE STUDY WAS CONDUCTED NEED TO BE DETAILED.
Ans. The details of the method section were already expanded on the revised version (Line 95-123).
- LINE 110.... T-scores were used to compare the study subjects to normalsex ... The meaning of this sentence is unclear. What women are matched by ethnicity?
Ans. Thank you for the question. The term ''sex''- and ''ethnicity''-matched individuals were used to describe ''female Caucasians'' in the reference database (NHANES/USA). Taiwanese and Caucasians belong to different ethnicity, but no significant difference was noted in fracture rate between Caucasians and Taiwanese. Therefore, diagnosis of osteoporosis based on WHO criteria, which is designed for white people originally, still can be used in postmenopausal women in Taiwan. We have added new reference in revised version. (Line 129-131)
- How many subject entered the multivariate model?
Ans: All 50 patients were entered the multivariate model.
- Line 120: SOMETHING ABOUT LABORATORY TESTS YOU USED SHOULD BE WRITTEN.
Ans. Details about biochemical measurements were now described at method section on revised version.
- Biochemical measurements
Biochemical measurements were performed using standardized methods in the central laboratory of our institution. Serum albumin, aspartate aminotransferase (GOT), alanine aminotransferase (GPT), thyroid stimulating hormone (TSH), free T4, T3, thyroglobulin, thyroglobulin and microsomal antibody, parathyroid hormone, total calcium, urine total protein and creatinine, serum creatinine , follicular stimulating hormone (FSH), estradiol, testosterone , cortisol , insulin growth factor-1 (IGF-1), bone alkaline phosphatase ( BALP), vitamin D and bone resorption marker carboxy-terminal collagen crosslinks ( CTX) were measured. (Line 116-123)
- Line 115: A T-score <-1 IS NOT NECESSARY abnormal. IT MAY BE NORMAL FOR AGE. Anyway It does not corresponds to the diagnosis of osteoporosis.
Ans: Thank you for raising this point. T-score is the standard deviation units used to evaluate the BMD status in relation to the young population. We agree that T-score < -1 may not necessarily corresponds to abnormal, but the diagnosis criteria of osteoporosis or low bone mass/osteopenia using T-score as reference is suggested to be used in postmenopausal women and in men aged 50 year and over. We also agree your opinion that T-score < -1 does not equal to diagnosis of osteoporosis. However, according to WHO criteria, T-score < -1 and > -2.5 represents low bone mass/osteopenia, which is also an abnormal status and the hallmark of progression to osteoporosis. Besides, all of our study individuals are postmenopausal women that eligible to use T-score as diagnosis reference. Thus we would like to retain T-score < -1 as cut-off point for “abnormal” in our study. References: Reference [27],[28],[29] in our manuscript.
- Table 1: weight and height are missing.
Ans: We have added height and weight in Table 1 as below.
|
Characteristics |
T-score ≥ -1.0 at each site (n = 12) |
T-score < -1 at any site (n = 38) |
p |
|
Height (cm) |
160.3 ± 2.7 |
155.3 ± 5.7 |
< 0.001 |
|
Weight (kg) |
63.9 ± 8.0 |
59.0 ± 9.2 |
0.085 |
- table 1: FEMORAL neck is g/cm2.
Ans: Thank you for your correction. We have revised this error.
- Table 3: THE T-SCORE IS A MEASURE DERIVED FROM THE BMD. GENERALLY THE BMD IS ONLY USED FOR THE STATISTICAL ANALYSIS OF THE DATA BECAUSE THERE SHOULD NOT BE STATISTICAL DIFFERANCES BETWEEN THE TWO MEASURES. HOW DO YOU EXPLAIN THE DIFFERENCE OF CORRELATION WITH TOTAL CALCIUM AND ASM BETWEEN BMD AND T-SCORE IN FEMORAL SITES?
Ans: BMD means the mean volumetric mineral density of body tissue, and T-score is standard deviation units in relation to the young healthy population. Although they are highly correlated, they are different markers. Therefore, their risk factors may be different.
- Line 261: You do not have selected women with osteoporosis. So your data suggest that sarcopenia is associated with BMD.
Ans. Thank you for your careful correction. We have replaced the term “osteoporosis” with “BMD” to make a more appropriate description.
- Line 230: BMD?
Thank you for your advice. We have added the finding about the relationship between IGF-1 and BMD in the text.
- But no significant association between IGF-1 and BMD was found. (Line 249-250)
- Women of the groups with "normal and abnormal" T-scores do not differ in nutritional status and muscle mass. Don't you think this contradicts the association of BMD with nutritional status and muscle mass? what is the usefulness of this comparison for the purpose of what the study wants to demonstrate?
Ans: We want to show the different characteristics of different T-score to the readers. Therefore, in table 1, we compare baseline and dual energy X-ray absorptiometry (DXA) characteristics between patients according to T-score ≥ or < -1 at each site. In addition, in table 1, T-score is used as category variable. However, in Table 2 and 3, T-score are use as continuous variable. Therefore, the results may be different.

Round 2
Reviewer 1 Report
The manuscript looks much better after the revision. The authors have made the suggested changes. As a result, the substantive value of the research has increased. However, it is not a good explanation for the low size of the study group that “Healthy postmenopausal women not usually visited our clinic, therefore we were not able to enroll this type of patient as control group". A group of healthy post-menopausal women is easy to recruit, for example through announcements of free analyses in health centers. It is good that the authors indicate that “Follow-up large-scale studies are needed to confirm our results".
Author Response
The manuscript looks much better after the revision. The authors have made the suggested changes. As a result, the substantive value of the research has increased. However, it is not a good explanation for the low size of the study group that “Healthy postmenopausal women not usually visited our clinic, therefore we were not able to enroll this type of patient as control group". A group of healthy post-menopausal women is easy to recruit, for example through announcements of free analyses in health centers. It is good that the authors indicate that “Follow-up large-scale studies are needed to confirm our results".
Ans: We apologized for our previous response was not able to meet your expectation. This time we try to contact our health management center, a center which has service includes the provision of hospitalized medical examination and medical healthy surveillance for intended individuals, which we think has the highest possibility to include healthy individual, and had undergone DEXA for osteoporosis evaluation with similar age group to our study population. However, we encountered the problems in the definition of “healthy” postmenopausal women. After surveillance of their medical history, maybe due to the aging, more than half of the surveyed patients has the comorbidities of the high risk association with osteoporosis like diabetes mellitus, chronic kidney disease, chronic liver disease included hepatitis B or C, or fatty liver, history of steroid user for pain, operation history for bone either traumatic induced or not, and etc. This history made us hesitate to include them to our study population as healthy control group. The less than half of the possible candidate as healthy control group, the will to join the study is low due to need to come back for additional permit signing, blood withdrawal and second round of DEXA for body composition study. Therefore, we have to apologize again and explain to you, your precious suggestion can raise the value of our manuscript, however, this time we need to put it in our limitation.

Reviewer 4 Report
Thank you to the Authors for their kind answers. Below is a list of my observations.
1) in line 96 you say that the study is prospective. It does not seem to me so.
2) Whether osteopenia or "low bone density" corresponds to abnormal bone mass state is your definition that is different from that of WHO (J Clin Endocrinol Metab 89: 3651–3655, 2004).
3) Are you sure that the T-score of the femoral neck BMD is correct (see table)?
4) My position is that T-score is perfectly correlated with BMD. A specific BMD value corresponds to a single T-score value. The correlation between BMD and T-score is biunivocal. If this were not the case, vitamin D, for example, would have to be correlated with BMD and not with the T-score. If the latter were in fact a measure not perfectly correlated with BMD, the relationship between the two direct measures of biological factors (i. E. Vitamin D status and BMD) would be altered by the correlation with the T-score.
5) Why are weight and height not entered in the multivariate? Are they also related to BMD?
6) About table 1. Since in the discussion you only deal with the correlation between BMD/T-score and the other parameters under study, I don't understand the usefulness of showing table 1. Furthermore, since nutritional status and muscle mass are directly related to BMD, the reader would expect find different nutritional status and muscle values among subjects with significantly different BMD subjects. But is not so.
Author Response
Thank you to the Authors for their kind answers. Below is a list of my observations.
1) in line 96 you say that the study is prospective. It does not seem to me so.
Ans: As your suggestion, we deleted the word “prospective”.
2) Whether osteopenia or "low bone density" corresponds to abnormal bone mass state is your definition that is different from that of WHO (J Clin Endocrinol Metab 89: 3651–3655, 2004).
Ans: Thank you for your comments. We agree with the reviewer’s assessment. The term “abnormal” is not used in terminology of WHO criteria. We use “abnormal” to described patient with BMD < -1, including those with osteopenia/low bone density and osteoporosis, in contrast with that BMD ≧ -1 is categorized as “normal” in WHO criteria. Besides, as a transitional phase from normal BMD to osteoporosis, osteopenia/ low bone density (T-score between -1 to -2.5) was also reported to be associated with increased fracture rate compared to those with “normal” bone density [A]. We had revised our Table 1 as presenting all patients.
[A] Zhang, Jie et al. “Osteopenia: debates and dilemmas.” Current rheumatology reports vol. 15,12 (2013): 384. doi:10.1007/s11926-013-0384-5
3) Are you sure that the T-score of the femoral neck BMD is correct (see table)?
Ans: Thank you for your careful correction. The corrected data was as below.
|
Characteristics |
T-score ≥ -1.0 at each site (n = 12) |
T-score < -1 at any site (n = 38) |
p |
|
T score |
0.03 ± 0.94 |
-2.16 ± 0.71 |
< 0.001 |
4) My position is that T-score is perfectly correlated with BMD. A specific BMD value corresponds to a single T-score value. The correlation between BMD and T-score is biunivocal. If this were not the case, vitamin D, for example, would have to be correlated with BMD and not with the T-score. If the latter were in fact a measure not perfectly correlated with BMD, the relationship between the two direct measures of biological factors (i. E. Vitamin D status and BMD) would be altered by the correlation with the T-score.
Ans: Thank you for raising this point. We totally agree with reviewer’s opinion. By definition, T-score is derived from BMD and a linear mathematical relationship could be found between T-score and BMD. However in our study, some minor differences are also noted between determinants of BMD and T-score (i.e. Bone ALP is the determinant of T-score but not BMD in Femoral neck in statistical analysis). We tried to examine the original data, and found some key-in errors. We have corrected the error, and re-analyzed our multivariable analysis. We found that IGF-1 was significantly associated with total hip BMD and T-score, not just T-score. However, total calcium and bone ALP were still just associated with femoral neck T-score, but not with femoral neck BMD. After consulting statistician in our university, we think that possible reasons are as follows.
- The data of BMD and T-score in our study are recorded directly from BMD report generated by DXA software. In the reports, the value of BMD is rounded off to 3rd decimal place, while the value of T-score is rounded off to 1st decimal place. Because of the relatively small value of T-score, this minor deviation from original value may interfere with the perfect linearity between BMD and T-score, and may further contribute to discrepancy between the result of determinants of BMD and T-score.
- T-score is a “weighted” index derived from BMD. Due to the impact of “weighted”, positive value of BMD may be shifted to negative value of T-score. The “direction” of unstandardized coefficient β would be the same for a given marker in BMD and T-score, however, the scale of the statistical analysis changed, then, the associated factors may be different between BMD and T-score.
5) Why are weight and height not entered in the multivariate? Are they also related to BMD?
Ans: Thank you for your question. BMD and T-score are corrected with age, body weight and race. Therefore, we did not put body mass index or weight into multivariable analysis for collinearity.
6) About table 1. Since in the discussion you only deal with the correlation between BMD/T-score and the other parameters under study, I don't understand the usefulness of showing table 1. Furthermore, since nutritional status and muscle mass are directly related to BMD, the reader would expect find different nutritional status and muscle values among subjects with significantly different BMD subjects. But is not so.
Ans: Thank you for your comments. We have revised our Table 1 as all patients.
